# On the Potential of Relational Databases for the Detection of Clusters of Infection and Antibiotic Resistance Patterns

**DOI:** 10.3390/antibiotics12040784

**Published:** 2023-04-19

**Authors:** Michela Gelfusa, Andrea Murari, Gian Marco Ludovici, Cristiano Franchi, Claudio Gelfusa, Andrea Malizia, Pasqualino Gaudio, Giovanni Farinelli, Giacinto Panella, Carla Gargiulo, Katia Casinelli

**Affiliations:** 1Department of Industrial Engineering, University of Rome “Tor Vergata”, 00133 Rome, Italy; 2Consorzio RFX (CNR, ENEA, INFN), University of Padua, 35127 Padua, Italy; 3Istituto per la Scienza e la Tecnologia dei Plasmi, CNR, 35100 Padua, Italy; 4Department of Biomedicine and Prevention, University of Rome “Tor Vergata”, 00133 Rome, Italy; 5ASL and Fabrizio Spaziani, Frosinone Hospital, 03100 Frosinone, Italy

**Keywords:** decision support system, infectious diseases, hotspot, relational databases, *Klebsiella*, nosocomial diseases, antibiotic resistance, antibiotic stockpiling

## Abstract

In recent years, several bacterial strains have acquired significant antibiotic resistance and can, therefore, become difficult to contain. To counteract such trends, relational databases can be a powerful tool for supporting the decision-making process. The case of *Klebsiella pneumoniae* diffusion in a central region of Italy was analyzed as a case study. A specific relational database is shown to provide very detailed and timely information about the spatial–temporal diffusion of the contagion, together with a clear assessment of the multidrug resistance of the strains. The analysis is particularized for both internal and external patients. Tools such as the one proposed can, therefore, be considered important elements in the identification of infection hotspots, a key ingredient of any strategy to reduce the diffusion of an infectious disease at the community level and in hospitals. These types of tools are also very valuable in the decision-making process related to antibiotic prescription and to the management of stockpiles. The application of this processing technology to viral diseases such as COVID-19 is under investigation.

## 1. Introduction

Robust and accurate knowledge of the frequency and geographic distribution of infections plays a key role in the reduction of disease contagion. Monitoring the spreading of antibiotic resistance represents an essential ingredient in counteracting the diffusion of microorganisms. To facilitate the surveillance system, it is mandatory to maintain easily updatable databases including all the cases of interest. Such a tool can be essential to verify any temporal and geographic trends in the spreading of the pathogen.

In this paper, we illustrate the potential of a relational database to provide essential information about the geographic localization of outbreaks and the multidrug-resistant properties of the bacterium *Klebsiellla pneumoniae* (*K. pneumoniae*) [1,2,3] in a specific region of central Italy (see Figure 1). The analysis is particularized for both external patients and nosocomial cases. The development and deployment of such a type of database is expected to become even more important in the post-COVID-19 era.

With regard to the structure of the paper, the rest of this section is devoted to the background on bacterial infections with particular attention to *Klebsiella pneumoniae*. Relational databases and the information processing technology deployed to perform the study of the present work are overviewed in Section 2. The results of the investigation are reported in detail in Section 3, while the conclusions are the subject of the last section of the paper. 

### 1.1. The Spread of Infections

Humans infected by a pathogen, depending on the state of the infection, may or may not transmit the pathogen. An individual who can transmit the pathogen is classified as a carrier [4,5] A passive carrier means a host that is carrying the pathogen can transmit it to another host, but that generally they are not infected [6]. For example, a healthcare professional who does not wash their hands properly after the examination of an infected patient could transmit the pathogen to another patient without becoming infected themselves [2]. On the contrary, an active carrier is an infected host that can directly transmit the infection but can show or not show the symptoms of the disease. Indeed, it can transmit the pathogen during the incubation period or in the recovery period [7,8,9]. Asymptomatic carriers are defined as those active carriers that do not show symptoms of the disease [10].

Pathogenic microorganisms, during their evolution, have developed different mechanisms for transmission. Among them, direct and indirect contact transmission represent two of the most diffused mechanisms. Person–person transmission is a form of direct contact transmission [1,11]. In this case, the agent is transmitted by physical contact between two individuals through actions such as touching, sexual intercourse, or droplet spread [12,13,14] Among the places that attract the interest of epidemiologists are hospitals and nursing settings due to high levels of disease transmission. Consequently, special efforts must be exerted to limit the risks of infection spreading in these environments. Infections acquired in healthcare facilities are called nosocomial or healthcare-associated infections (HAIs) [15,16] To classify an infection as an HAI, the patient must have been admitted to the healthcare facility for a reason other than the infection. In the case of nosocomial diseases, higher rates of transmission may be caused by characteristics of the environment itself, characteristics of the population, or both [17]. HAIs are often connected with surgery or other invasive procedures that facilitate access of the pathogen to the host [18,19] Indeed, in these settings, patients suffering from a primary disease are often afflicted by compromised immunity and are more susceptible to secondary infections and opportunistic pathogens [7,8].

### 1.2. Klebsiella Pneumoniae

*Klebsiella pneumoniae* is a non-motile, Gram-negative, lactose fermenting, rod-shaped, and encapsulated bacterium. It is widely diffused in the world, mainly in tropical and sub-tropical areas [20]. *K. pneumoniae* is the most clinically relevant member of the *Klebsiella* genus, a member of *Enterobacteriaceae* family [21,22,23,24]. *Klebsiella* bacteria are ubiquitous and probably have two common habitats, one is the natural environment, where they are found in surface water, soil and plants. The other habitat is represented by the mucosal surfaces of mammals such as humans, horses, and swine [25]. In humans, *K. pneumoniae* is present as a saprophyte in the nasopharynx and in the intestinal tract.

Normally, individuals carry *K. pneumoniae* asymptomatically [26,27]. However, in immunocompromised individuals, especially in infants and elderly, *K. pneumoniae* can cause severe hospital-acquired infections [28,29,30] such as pneumonia and urinary tract infections. In addition, certain hypervirulent *K. pneumoniae* strains can cause community-acquired infections [10,31,32]. The acquisition of antibiotic resistance genes and intrinsic resistance to several classes of antibiotics limit treatment options for infections caused by this [33,34,35].

Currently, *K. pneumoniae* strains producing extended spectrum beta-lactamases (ESBLs) and carbapenemases have spread globally [36,37]. These types of enzymes inactivate β-lactams, a class of antibiotics that is normally the basis for effective treatment of patients suffering from infections with *K. pneumoniae*. Consequently, the antimicrobial peptide (AMP) and colistin (also known as polymyxin E) have been reintroduced to treat infections with multidrug-resistant *K. pneumoniae* [13]. The rise of these multidrug-resistant isolates will require the development of novel classes of antibiotics or alternative treatment strategies.

### 1.3. Antibiotic Resistance

The treatment of bacterial infections depends heavily on effective antimicrobial therapy. The combination of the wide use of inappropriate therapies and the delayed administration of effective antibiotics have been associated with a higher mortality rate in patients with severe infections (especially nosocomial patients) due to the diffusion of resistances [38]. For these reasons, nosocomial ESBL-producing *K. pneumoniae* infection has been considered an emerging threat. *K. pneumoniae* is a major cause of nosocomial infections and is responsible for roughly 15% of Gram-negative infections in hospital intensive care units (ICUs), primarily affecting immunocompromised patients [35,39]. In these cases, the presence of drug-resistant pathogens would adversely affect the treatment outcome [15].

Data on healthcare-associated infections reported by the Centers for Diseases Control and Prevention (CDC) from 2007 indicates that the overall prevalence of ESBL-producing *K. pneumoniae* isolates varies considerably depending on the geographic region [39,40]. According to the CDC, the geographical distribution of ESBL-producing *K. pneumoniae* is as follows: Eastern Europe (39.3%), Korea (26.2%), USA (16%), and Canada (3.6%) [17,40,41,42].

In Italy, as reported by the Istituto Superiore di Sanità (ISS), Gram-bacteria have shown increased levels of resistance during the period 2006–2008, especially to specific classes of antibiotics (such as fluoroquinolones, aminopenicillin, aminoglycoside, and carbapenems). From 2009 to 2011, a worrying trend was registered where the percentage of strains of *K. pneumoniae* resistant to carbapenems presented a 1.3% increase in 2009, a 16% increase in 2010, and a 26.7% increase in 2011. In order to stop this escalation, it is, therefore, necessary to pay particular attention to optimizing antibiotic therapies and to rationalizing antibiotic-sensitivity tests [43,44,45,46,47].

### 1.4. The Role of Prevention

To reduce the diffusion of infectious diseases, prevention plays a key role [47]. By monitoring the nosocomial frequency and the geographic distribution of invasive infections due to multidrug-resistant organisms (MDROs), specific strategies can be developed in order to implement prevention procedures. Special tools are needed to collect and review the cases and to quantify the actual risk of infection spread [48,49,50,51,52].

One of the most effective tools, perhaps the most important, is the judicious use of antimicrobial agents applicable to both acute and long-term care settings [53,54]. In many studies, it has been observed that changing a certain antimicrobial therapy is associated with the disappearance of an MDRO target. However, single and unique antimicrobial therapies are not enough as multidrug resistance is caused by a number of combined factors, including initial selective pressure, perpetuation of resistance, inappropriate use, or insufficient time to observe the effect of the therapy [17]. Furthermore, as part of an antimicrobial stewardship program, facilities should work to ensure that antimicrobials are administered according to appropriate indications and duration and that the narrowest spectrum antimicrobial that is appropriate for the specific clinical scenario is used [25].

## 2. Databases Technology and Content

This section describes the main aspects of the relational database technology (Section 2.1) and gives an overview of the database built for the investigation, which is the subject of the present paper (Section 2.2).

### 2.1. Relational Databases as Decision Support Systems

Relational databases became the dominant model in the 1980s. The term “relational database” was defined by E. F. Codd at IBM in 1970 [55]. The formal and most complete definition of what constitutes a relational database is based on Codd’s 12 rules. However, no commercial implementations of the relational model conform to all of Codd’s requirements [56]. The term has, therefore, gradually been used to identify a broader class of database systems which, at a minimum, have to present the following characteristics:

Presentation of the data to the user as relations: the presentation format is tabular, i.e., consists of a collection of tables with each table comprising a set of rows and columns.

IBM began developing System R, a research project aimed at developing a prototype relational database management system (RDBMS) in 1974. The first software package commercialized as an RDBMS was Multics Relational Data Store (June 1976). The first version of Oracle was released in 1979 by Relational Software, which is now the Oracle Corporation. 

Relational databases are organized according to the relational model, which consists of an intuitive way of representing data in tables. In practically all implementations, each row in the table of a relational database is a record with a unique ID called the key. The columns of the table contain the attributes of the data where, in principle, each record usually holds a specific value for each attribute. Thanks to its logical structure and practical implementation, a relational database allows the easy finding specific information. The retrieved information can be easily sorted on the basis of any field from each record. Moreover, it is easy to compare information thanks to the arrangement of data in columns. Relational databases are built using a specific computer language, called structured query language (SQL), which is the established standard for database interoperability.

The power and flexibility of relational databases has become very valuable to support the decision-making process in the sanitary or healthcare context at different levels. First, they can help significantly in the phase of monitoring the general situation and, in the specific case of the spread of infectious diseases, in detecting hotspots and outbreaks in a timely manner. Relational databases can also provide extremely useful information to organize interventions and prepare for them. In the present application, for example, they can be a very effective basis for managing the storage and reserves of antibiotics. These software tools have also great potential in providing support to the organization of prevention strategies. Most of these aspects will be exemplified by the analysis of the database described in the following subsection.

The software used to analyze the data presented in this work was the package ACCESS (Microsoft^®^ Access^®^) and Microsoft^®^ Excel^®^.

### 2.2. The Database

Anonymized data were provided by A.S.L. Frosinone (the local Healthcare Service of Frosinone city) for the region of central Italy shown in Figure 1. It covers the period that spanned from January 2014 to March 2016. For each patient, the available information was progressive code, acceptance date, initials of the name, date of birth, sex- origin, material, bacterial charge, results of antibiogram, and antibiograms.

The antibiogram test utilizes a variety of culture methods which work by exposing bacteria to antibiotics. Culture methods typically consist of measuring the diameter of areas without bacterial growth, the so-called zones of inhibition, around paper discs containing antibiotics on agar culture dishes. The culture dishes had been previously inoculated evenly with bacteria. The lowest concentration of an antibiotic which stops the growth of bacteria is the minimum inhibitory concentration and can be determined by observing the size of the zone of inhibition.

The blood samples of the patients were tested to assess the resistance to the following antibiotics ampicillin, amoxicillin, piperacillin, cefoxitin, ceftazidime, cefepime, ertapenem, imipenem, meropenem, tigecycline, fosfomycin, cyprofloxacin, amikacin, gentamicin, and colystin. To quantify the level of resistance, three classes were defined: (1) sensitive S, (2) intermediate I, and (3) resistant R. A bacterial strain is classified as susceptible to a certain antibiotic when it is inhibited in vitro by a concentration of the drug which corresponds to a high likelihood of therapeutic success. The sensitivity of a bacterial strain to a given antibiotic belongs to the intermediate category when it is inhibited in vitro by a drug concentration which presents an uncertain therapeutic effect. A bacterial strain is categorized as resistant to a given antibiotic when it is inhibited in vitro by a drug concentration, which tends to result in therapeutic failure.

## 3. Results

This section is devoted to an overview of the obtained results for the database described in Section 2.2. As mentioned in the abstract and in the introduction, the main purpose of the present work consisted of showing the potential of relational databases in supporting the decision-making process at the beginning and during an epidemic. These tools can indeed help in detecting the hotspots of the infection and optimizing countermeasures, both in the community and in care institutions. To illustrate these capabilities, the section is organized as follows. The global statistics about the whole database are reported in Section 3.1. Detailed information about the external and internal patients is provided in Section 3.2 and Section 3.3, respectively.

### 3.1. Global Statistics

A schematic view of the province and its municipalities, from which the patients originated from, is provided in Figure 1. The data covered the period between January 2014 and March 2016. In this interval, up to 665 patients were found to be affected by *K. pneumoniae*. Of these, 594 were external (75% of the total) and 171 were internal (25% of the total). 

Of the external patients, 348 were women and 146 were men. Females amounted for 91 cases and males 75 cases for the internal patients (in five entries, the sex of the patient was missing). Overall, the average age of the patients was 63 years and 61% of them were women and the remaining 39% men. Regarding age, 63% were more than 65 years old, 15% had an age between 45 and 64, 9% between 25 and 34, 3% between 4 and 24, and only 1% were in the interval between 0 and 3.

Figure 2 shows the number of cases versus the age of the patients. A trend with age is clearly seen. Particularly affected are the individuals older than 64 years of age.

The antibiotic resistance of all the patients in the database is summarized in the plot of Figure 3. To investigate the evolution of infection and the multidrug resistance of the antibiotics in more detail, it is appropriate to distinguish two classes of patients, the external ones and the internal ones, who have contracted the disease after hospitalization. 

### 3.2. Statistics of Provenance and Antibiotic Resistance for External Patients

One of the main aims of this study is to show that the proposed decision support system can help in identifying hotspots of infection in the territory. For the external patients, two characteristics have a particular importance: (a) the geographical provenance and (b) the multidrug resistance.

The distribution of the infections per province of the patients is shown in the pie chart of Figure 4. The highest incidence, in percentage of the population, is detected in the province of Ceprano, where 1% of the overall residents were affected. Ceccano is the second with 0.2%, Boville the third with 0.15%, and Ferentino the fourth with 0.1% of the population infected by *K. pneumoniae*.

Due to missing data, for the detailed investigation of the multidrug resistance it was necessary to limit the detailed analysis of the external patients to the period between April and September 2014, which in any case is the time interval of the largest number of cases. In this period, the sample was made of 113 external patients.

Figure 5 reports the percentage of resistance to individual antibiotics for the external patients, from which it can be seen that, as expected, 99% of the patients do not respond to ampicillins. On the other hand, the percentage resistance to carbapenems, tigecycline, and aminoglycosides remains quite low.

Table 1 reports the combined resistances between antibiotics. To complete the overview of the available data, the combined resistances to the various antibiotics per province of origin of the patients is reported in Table 2.

To summarize, Frosinone and Ceccano were the municipalities most affected by *K. pneumonia* in absolute terms. The same municipalities were also the only ones in which patients showed resistance to more than six families of antibiotics. In general, the vast majority of cases of MDR were also detected only in the districts of Ceccano and Frosinone. It is, therefore, reasonable to assume that some particularly resistant strains of the bacterium evolved in these municipalities.

In general, with regard to the MDR, penicillins (particularly ampicillin) and the celaphosporines (particularly cephioxitin) are not very effective against *K. pneumoniae*. This evidence is probably related to the abuse of antibiotics, which tends to reduce the immune systems and at the same time favors the evolution of resistant variants.

The diffusion of carbapenemase-producing enterobacteriacae (CPE) and *Klebsiella pneumoniae* carbapenemase (KPC) bacteria is increasing significantly, which is a cause of concern because these pathogens are known to show a higher resistance to the antibiotic families under observation by the ECDC: carbapenemes, gliciliclines, and poliximines. Fortunately, and at least until the date of present survey, glycylcyclines and polymyxins remain quite effective. Moreover 93 % and 90 % of the cases resistant to gliciliclines and poliximines are also resistant to a maximum of two other antibiotics. Only 3% and 4%, respectively, are also resistant to more than other six antibiotic families, but these were all subjects of 60 years of age or older. With regard to the carbapenemes, the resistance levels are also quite low; 3% for the imipenem and 4% for the meropenem, respectively. Only for the ertapenem was a higher level of 16% found. In general, for the entire carbapeneme family, 83% of the resistant patients have shown resistance to a maximum of two other antibiotics. To summarize, the gliciliclines and poliximines, and to a lesser extent the carbapenemes, remain the most effective antibiotics against the pathogen *K. pneumoniae.* In general, MDR patients were older than 60 years of age. 

### 3.3. Statistics of Nosocomial Cases

The quality of the data for the internal patients is typically higher. A complete set of data is available for 152 patients. The distribution of the nosocomial cases for department is reported in the pie chart of Figure 6. 

The highest number of patients contracting infections inside the hospital occurs around summer.

Since one of the main objectives of the present work is to show the potential of the software tools for the investigation of antibiotic resistance, the attention has been focused on the patterns of multidrug resistance. Only 39% of the patients do not share any form of resistance pattern. All the other 61% belong to 18 patterns identified with the help of the developed software tools. The histogram of Figure 7 reports the number of cases for each of the detected patterns. Non-effective antibiotics are shown in red, the ones with a good efficacy are in green, and the color yellow is used to indicate the intermediate situations. The multidrug resistance for each of the seven patterns, containing at least five patients, is reported in Table 3.

In 34 cases out of 93 (37%), the bacterium show a resistance to more than 12 antibiotics. The only antibiotic that is effective against all the strains of the bacterium is the colistin, which, together with the gentamicin, is very toxic. These data confirm the need to find new antibiotics. Indeed, a study conducted in nine hospitals in Rome between 2010 and 2011 discovered *Klebsiella* strains resistant to colistin in 36% of the cases (the resistance to gentamicin was recorded in 80% of the cases). Resistance to colistin is a fundamental cause of mortality [54,57].

The patterns of MDR for the two most affected wards, intensive care and infectious diseases, are reported in the histograms of Figure 8.

## 4. Discussion and Conclusions

The WHO [58] has identified 12 families of bacteria as the most dangerous for human health with the objective of promoting research into new antibiotics. These bacteria are divided into three categories and *Klebsiella* spp. is included in the first, the one considered potentially most harmful. In the period between 2008 and 2011, according to the ECDC, 15,052 cases of *Klebsiella pneumoniae* were detected in Europe. In terms of geographical distribution, 2.3% of these cases were detected in Sweden and 81% in Bulgaria, the least and the most affected countries, respectively. Unfortunately, a decrease of the resistance to the cephalosporins has been observed in no country. In Italy, the Ministry for Health has reported a significant trend of increased drug resistance. In 2009 only 1.35% of the *Klebsiella pneumoniae* strains showed antibiotic resistance, and this percentage increased to 16% in 2010 and to 26.7% in 2011.

Antibiotics are the main weapon against the diffusion of bacterial infections. They are also the most used class of drugs, with a high impact on the budget of any healthcare service. The abuse of antibiotics and the consequent reduction of their efficacy is, therefore, a very significant problem. The present study confirms that in Italy the diffusion of Gram-negative bacteria of the *Klebsiella pneumoniae* family and those resistant to the carbapenems is increasing.

The tools presented in this work have the potential to significantly help in the efforts to counteract the spread of multidrug-resistant bacteria. As shown, they can be very effective in helping to identify hotspots early, both in the territory and inside hospitals. Relational databases can also provide timely information about the evolution of MDR infections for more prompt reactions.

Relational databases such as that one shown in the present work can also help significantly in the management and procurement of the stockpiles of antibiotics, with consequent appreciable reduction in costs. In the case of the internal patients analyzed in the present study, the hospitalization lasted 11 days on average for the 39% MDR patients and only 5 for the others. Considering an average cost of about 1000 euros per patient per day of hospital stay, the MDR patients more than doubled the costs of hospitalization. For any reasonable economic burden required to maintain the software tools proposed in this paper, the savings per year of reducing MDR resistance would be very significant. In terms of antibiotics procurement, the savings would be in the order of 30%. The previous estimates refer to *Klebsiella* bacteria only and do not include the potential reduction in costs for personnel and storage areas. The capability of relational databases to improve the quality of care in all respects is, therefore, obvious and important.

Of course, and as it is typically the case of all sophisticated software tools, relational databases have to be properly maintained and updated. Conversely, they can easily become useless or even counterproductive. However, once developed, the costs of keeping such a database up to date should be minimal. To ensure their effectiveness, however, data collection has to be devoted proper attention. In the present relatively small database, the errors in the entries were also significant. Data collection and curation are typically the most delicate and resource intensive activities in the development of these information processing tools.

## Figures and Tables

**Figure 1 antibiotics-12-00784-f001:**
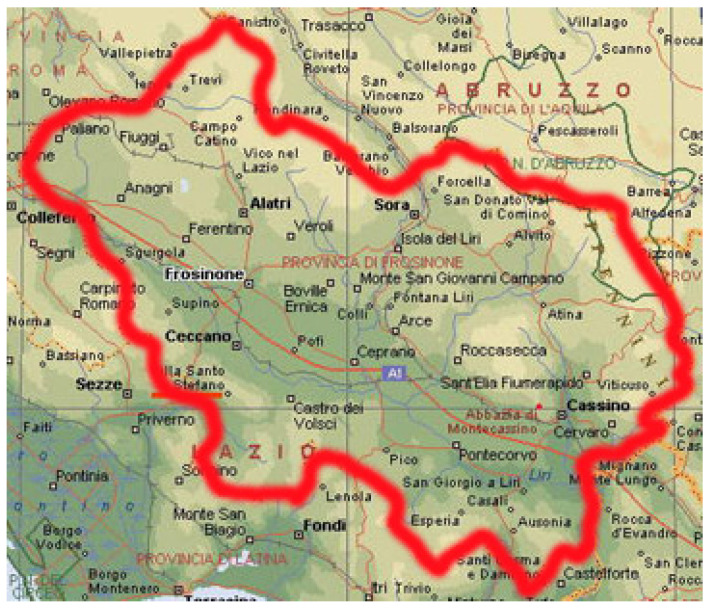
Provinces and municipalities of central Italy from which the patients affected by *K. pneumoniae* and included in the database (see Section 2.2) came from.

**Figure 2 antibiotics-12-00784-f002:**
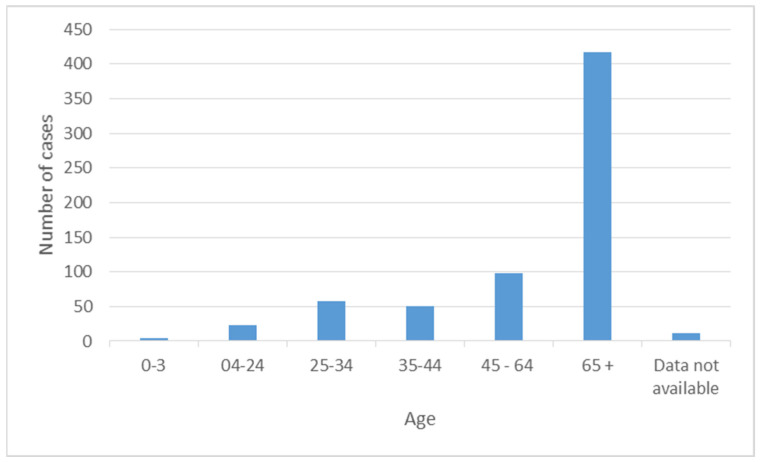
Number of cases versus age for the patients affected between January 2014 and March 2016.

**Figure 3 antibiotics-12-00784-f003:**
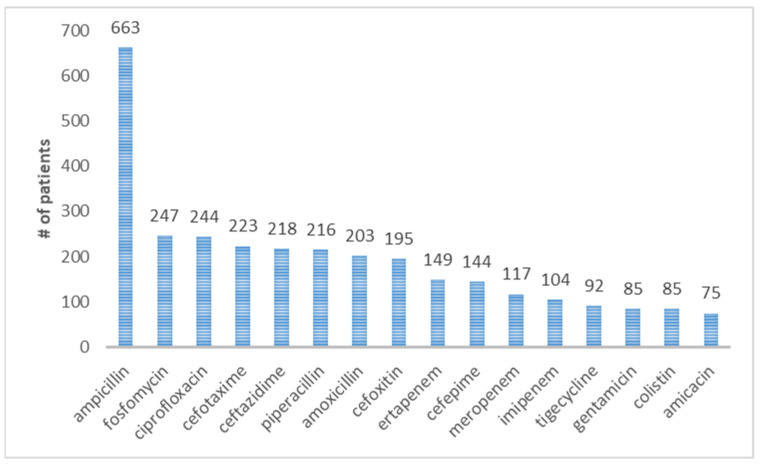
Antidrug resistance of the *K. pneumoniae* strains isolated from patients in the database described in Section 2.2.

**Figure 4 antibiotics-12-00784-f004:**
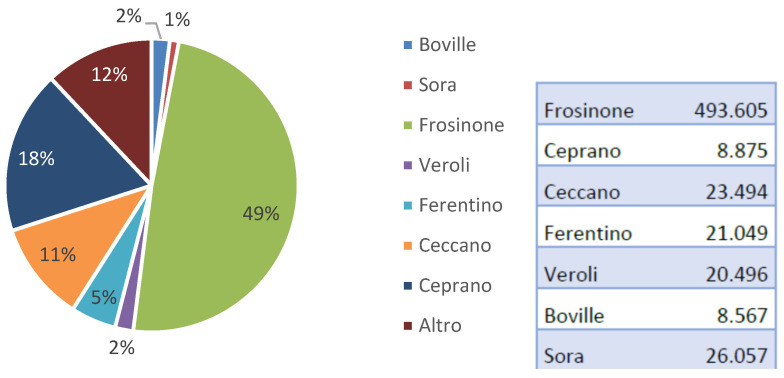
Left: Distribution of the patients’ provenance in percentage of the 494 external patients. Right: 2017 population of the various provinces as provided by the Italian National Institute of Statistics. Data were sourced from the database described in Section 2.2.

**Figure 5 antibiotics-12-00784-f005:**
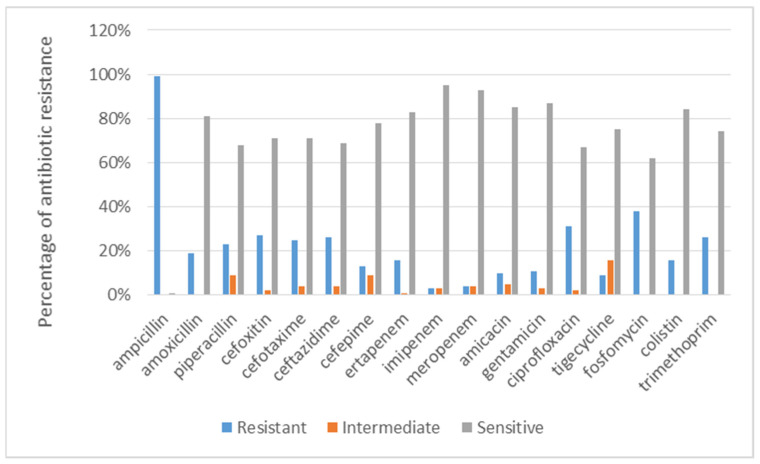
Percentages of antibiotic resistance for external patients in the database described in Section 2.2.

**Figure 6 antibiotics-12-00784-f006:**
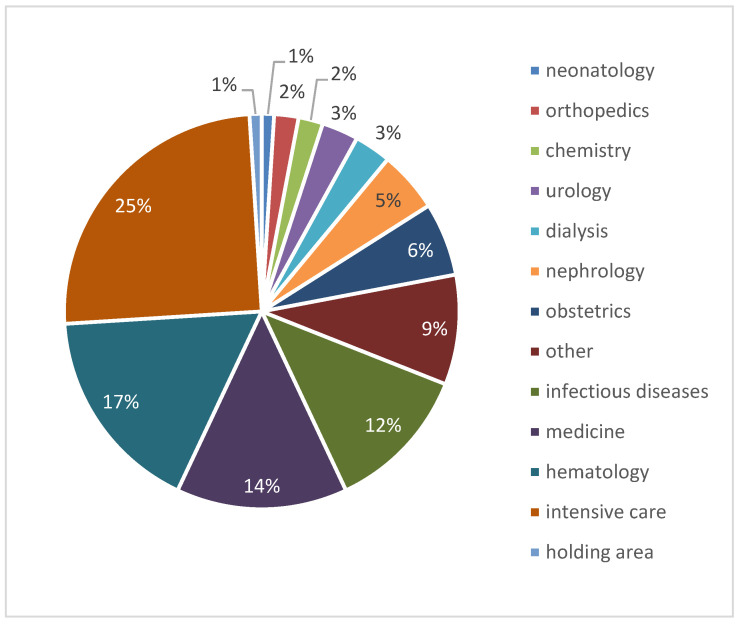
Distribution of nosocomial cases as a function of hospital department (from the database described in Section 2.2).

**Figure 7 antibiotics-12-00784-f007:**
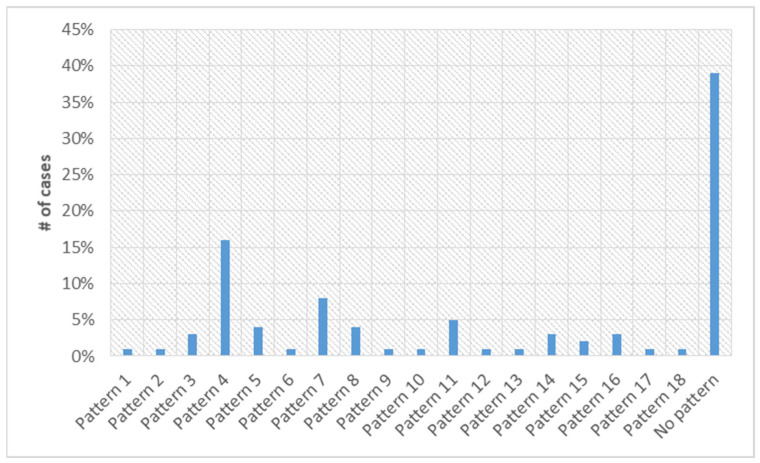
Number of cases for each of the 18 patterns of multidrug resistance for the 152 nosocomial patients (from the database described in Section 2.2).

**Figure 8 antibiotics-12-00784-f008:**
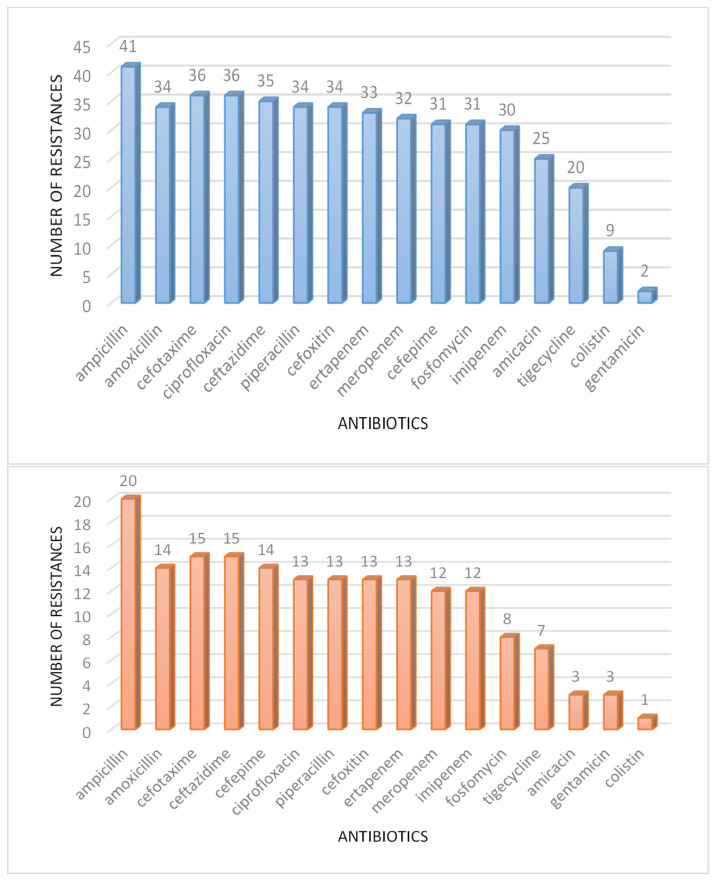
MDR resistance patterns in intensive care (upper plot, 42 cases) and infectious diseases (under plot, 20 cases) wards (from the database described in Section 2.2).

**Table 1 antibiotics-12-00784-t001:** Multidrug resistance. The level of combined resistance to couples of families of antibiotics in the period between April and September 2016 (from the database described in Section 2.2).

	Penicillins		Cephalosporins		Carbapenems		Aminoglycosides		Fluoroquinolones		Glycylcycline		Fosfomycin		Polymyxins		Diaminopyridines	
penicillins			45	40%	19	17%	20	31%	35	31%	10	9%	40	35%	16	14%	28	25%
cephalosporins	45	40%			18	17%	17	15%	30	27%	8	7%	24	21%	11	10%	18	16%
carbapenems	19	17%	17	15%			7	6%	13	12%	4	4%	12	11%	9	8%	11	10%
aminoglycosides	20	18%	17	15%	7	6%			17	15%	4	4%	13	16%	5	4%	10	9%
fluoroquinolones	35	31%	30	27%	13	12%	17	15%			6	5%	18	16%	7	6%	20	18%
glycylcycline	10	9%	8	27%	4	4%	4	4%	6	55%			8	7%	4	4%	2	2%
fosfomycin	40	35%	24	21%	12	11%	13	12%	18	16%	8	7%			9	8%	18	16%
polymyxins	16	14%	11	10%	9	8%	5	4%	7	6%	4	4%	9	8%			6	5%
diaminopyridines	28	25%	18	16%	11	10%	10	9%	20	18%	2	2%	18	16%	18	16%		

**Table 2 antibiotics-12-00784-t002:** Province of origins of the multi-resistant external patients in the period between April l and September 2016 (from the database described in Section 2.2).

Cobined Refreshments	Antibiotics	Frosinone (*p* = 66)		Veroli (*p* = 6)		Boville (*p* = 3)		Ceprano (*p* = 21)		Ferentino (*p* = 3)		Ceccano (*p* = 14)	
0	penicillins	41	62%	3	50%	1	33%	17	81%	2	67%	13	93%
	cephalosporins	42	64%	3	50%	1	33%	17	81%	2	67%	13	93%
	carbapenems	50	76%	6	100%	3	100%	20	95%	2	67%	13	93%
1	aminoglycosides	53	80%	3	50%	2	67%	19	90%	3	100%	13	93%
	fluoroquinolones	46	70%	3	50%	1	33%	18	86%	3	100%	13	93%
	glycylcycline	61	92%	6	100%	3	100%	20	95%	2	67%	13	93%
2	fosfomycin	50	76%	5	83%	1	33%	18	86%	3	100%	13	93%
	polymyxins	59	89%	6	100%	3	100%	19	90%	2	67%	13	93%
	diaminopyridines	49	74%	5	83%	1	33%	19	90%	3	100%	14	100%
3	penicillins	20	30%	3	50%	2	67%	3	14%	1	33%	0	0%
	cephalosporins	18	27%	3	50%	2	67%	2	10%	1	33%	0	0%
	carbapenems	11	17%	0	0%	0	67%	0	0%	1	33%	0	0%
6	aminoglycosides	9	14%	3	50%	1	0%	1	5%	0	0%	0	0%
	fluoroquinolones	15	23%	3	50%	2	33%	1	5%	0	0%	0	0%
	glycylcycline	3	5%	0	0%	0	67%	0	0%	1	33%	0	0%
5	fosfomycin	10	15%	1	17%	2	0%	2	10%	0	0%	0	0%
	polymyxins	6	9%	3	0%	0	67%	1	5%	1	33%	0	0%
	diaminopyridines	11	17%	1	17%	2	0%	2	10%	0	0%	0	0%
6	penicillins	5	8%	0	0%	0	0%	1	5%	0	0%	1	7%
	cephalosporins	6	9%	0	0%	0	0%	2	10%	0	0%	1	7%
	carbapenems	5	8%	0	0%	0	0%	1	5%	0	0%	1	7%
7	aminoglycosides	4	6%	0	0%	0	0%	1	5%	0	0%	1	7%
	fluoroquinolones	5	8%	0	0%	0	0%	2	10%	0	0%	1	7%
	glycylcycline	2	3%	0	0%	0	0%	1	5%	0	0%	1	7%
8	fosfomycin	6	9%	0	0%	0	0%	1	5%	0	0%	1	7%
	polymyxins	1	2%	0	0%	0	0%	1	5%	0	0%	1	7%
	diaminopyridines	6	9%	0	0%	0	0%	0	0%	0	0%	0	0%

**Table 3 antibiotics-12-00784-t003:** Graphical representation of multidrug resistance. The antibiotics that are not effective are red, the ones with a good efficacy are green, and the intermediate are yellow (from the database described in Section 2.2).

Antibiotics/Pattern	Pattern 4	Pattern 7	Pattern 11	Pattern 5	Pattern 8	Pattern 3	Pattern 16
Ampicillin							
Amoxicillin							
Piperacillin							
Cefoxitin							
Cefotaxime							
Ceftazidime							
Cefepime							
Ertapenem							
Imipenem							
Meropenem							
Amicacina							
Gentamicin							
Ciprofloxacin							
Tigecycline							
Fosfomycin							
Colistin							

## Data Availability

Data available on request due to restrictions.

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
