# Peer review of "On the Potential of Relational Databases for the Detection of Clusters of Infection and Antibiotic Resistance Patterns"

_antibiotics, 2023, doi:10.3390/antibiotics12040784_

Round 1
Reviewer 1 Report
Gelfusa et al. illustrated relational database to provide information on multidrug-resistant Klebsiella pneumoniae in specific region of Italy. While the paper is interesting, poor organization and writing of the manuscript has hampered my understanding of the findings.
Firstly, objective of this study was not clear. From my understanding, the authors tried to develop database on MDR Klebsiella in their study but I could not seem to find the database in the results. Abstract was written poorly and Introduction section was too long. Section 2.1 was also written too long.
Secondly, the results did not reflect the objective, in which they were mainly on epidemiology description. Again, the way they were written hampered my understanding to see relationship between objective and result.
Thirdly, the way this paper was written made me wonder whether it is a review or original article. Perhaps the authors can rewrite this paper with better organization.
Lastly, conclusion was too long and what was the relevance of including tables in conclusion?
Author Response
Review Report Form
Gelfusa et al. illustrated relational database to provide information on multidrug-resistant Klebsiella pneumoniae in specific region of Italy. While the paper is interesting, poor organization and writing of the manuscript has hampered my understanding of the findings.
ANSWER: we would really like to thank the reviewer for the time and effort spent revising and correcting our manuscript. We are also sorry to hear that the reading was difficult and that we did not manage to convey a clear message. We have therefore reread the entire manuscript and tried to improve the logic of the exposition to the best of our abilities. Of course we remain available to implement further modifications if the reviewer wanted to send us some more specific indications about possible improvements.
Firstly, objective of this study was not clear. From my understanding, the authors tried to develop database on MDR Klebsiella in their study but I could not seem to find the database in the results.
ANSWER: the objective of the work should be clear in the new version of the abstract. We have emphasised the point also in the introduction: “In this paper we illustrate the potential of a relational database to provide essential information about the geographical localisation of outbreaks and the Multi-Drug Resistant properties of the bacterium Klebsiellla pneumoniae (K. pneumoniae) in a specific region of central Italy. The analysis is particularised for both external patients and nosocomial cases. The development and deployment of such a type of databases is expected to become even more important in the post COVID-19 era”. Coming to the data, an overview of the database is provided in subsection 2.2 and the derived global statistics are summarised in subsection 3.1. More detailed analysis is the subject of the rest of section 3. We have specified that the results obtained refer to the database collected in a central region of Italy and described in subsection 2.1. We are pretty sure that the journal did not make any particular request about the availability of the raw data at the moment of the submission. We are available to provide the reviewer with the original data but we would have to ask her/him to sign a nondisclosure agreement, because some of the entries are sensitive (and it would certainly take some time to prepare such a document).
Abstract was written poorly and Introduction section was too long. Section 2.1 was also written too long.
ANSWER: we have shortened the introduction significantly even if we did not have a lot of room for manoeuvring, because the other reviewer was happy with these parts and even asked for some more details. We did not modify section 2.1, because it describes quite concisely the main tool used in the work, relational databases. We have tried to improve the abstract but we would need more specific indications by the reviewer to do a better job.
Secondly, the results did not reflect the objective, in which they were mainly on epidemiology description. Again, the way they were written hampered my understanding to see relationship between objective and result.
ANSWER: we have tried to clarify the logic of the manuscript. In particular we have emphasised that the objective of the work consists of exemplifying the usefulness of relational databases for the investigation and management of epidemic diseases, both in society and in hospital settings (see previous answer).
Thirdly, the way this paper was written made me wonder whether it is a review or original article. Perhaps the authors can rewrite this paper with better organization.
ANSWER: We have tried to improve the exposition, compatibly with the short time available for the review (only 10 days), which is not sufficient for a complete rewriting of the work. In addition to the new abstract and part of the introduction, the new beginning of section 3 should be now sufficiently explicit: “This section is devoted to the overview of the obtained results for the database described in Subsection 2.2. As mentioned in the abstract and in the introduction, the main purpose of the present work consists of showing the potential of relational databases in supporting the decision making process at the beginning and during an epidemic. These tools can indeed help detecting the hotspots of the infection and optimising the countermeasures, both in the community and in care institutions. To illustrate these capabilities the section is organised as follows. The global statistics about the whole database are reported in subsection 3.1. The information about the external and internal patients is provided in subsections 3.2 and 3.3 respectively”.
Lastly, conclusion was too long and what was the relevance of including tables in conclusion?
ANSWER: again we have shortened slightly the conclusions. However we thought that the discussion there was quite interesting, to put the local results in a more general national and European context. We have therefore modified the title of the last session, which is now called Discussion and conclusions. With regard to figure 8 and table III, they are in the conclusions (section 4) only due to the layout of the manuscript. They are indeed both referenced in section 3 on the results. Together with the editor we’ll try to find a better page setting without these elements in the concluding section.
Reviewer 2 Report
Dear authors, Individual comments and remarks are marked in the form of comments directly on the file. To sum up, the topic of the publication is very interesting and the most up-to-date in terms of growing and spreading antibiotic resistance sodium bacteria. However, the microbiological data themselves are insufficiently detailed and correctly described and presented.

Author Response
Review Report Form
Dear authors, Individual comments and remarks are marked in the form of comments directly on the file. To sum up, the topic of the publication is very interesting and the most up-to-date in terms of growing and spreading antibiotic resistance sodium bacteria. However, the microbiological data themselves are insufficiently detailed and correctly described and presented.
ANSWER: first we would really like to thank the reviewer for the time and effort spent reading and correcting our manuscript. We are also very grateful for the encouraging words and the positive comments. We have tried our best to implement all the corrections and suggestions. The modifications have been made with word tracking but we summarise the main changes in the following for the reviewer convenience:
- We have clarified that the figures and results refer to the database covering the central region of Italy shown in figure 1.
- We have improved the notation following the reviewer’s indications.
- We have included additional references as suggested by the reviewer.
- We have deleted the last sentence in the conclusions, which tried to make a parallel between bacterial and viral epidemics.
One last remark. The database does not contain any information about the NDM phenotype. Unfortunately we would not know how to obtain it, particularly in the short time allowed for the revision (10 days). Our understanding is that routine blood tests do not include the NDM phenotype and we wanted to analyse a large database of normally available information. Moreover, we believe that this detail is probably not crucial for the goal of the work, which consists of proving the usefulness of the relational database technology. We’ll certainly keep this aspect in mind when building new datasets for future publications.
Round 2
Reviewer 1 Report
Thank you for addressing my comments.
The authors have already addressed my comments.
However Figures and Tables in Result section are not organized properly.
Figure 1 and 2 should be arranged next to the text.
Y-axis of Figure 3 needs to be labeled.
Author Response
Thank you for addressing my comments.
The authors have already addressed my comments.
However Figures and Tables in Result section are not organized properly.
ANSWER: we have revised the layout of the manuscript and we’ll interact with the editorial office to optimise it for the final version.
Figure 1 and 2 should be arranged next to the text
ANSWER: we have revised the layout of the manuscript and we’ll interact with the editorial office to optimise it for the final version.
Y-axis of Figure 3 needs to be labeled.
ANSWER: many thanks for spotting the issue. We have added the label to the Y axis
Reviewer 2 Report
Akceptuje po dokonanych poprawkach.